# Immobilization of *Fusarium solani* Cutinase onto Magnetic Genipin-Crosslinked Chitosan Beads

**Zhanyong Wang** [1,*] **, Tingting Su** [2,*] **and Jingjing Zhao** [2]

1   College of Bioscience and Biotechnology, Shenyang Agricultural University, Shenyang 110866, China
2   School of Petrochemical Engineering, Liaoning Petrochemical University, Fushun 113001, China; jingjing6180@126.com
*   Correspondence: wangzy125@syau.edu.cn (Z.W.); sutingting@lnpu.edu.cn (T.S.); Tel.: +86-24-8848-7163 (Z.W.); +86-24-5686-1705 (T.S.)

**Abstract:** Genipin was used as a crosslinking agent to prepare magnetic genipin-crosslinked chitosan beads, which were then used as a carrier for immobilizing recombinant cutinase from *Fusarium solani* (FSC) to obtain immobilized FSC. The optimal temperature for the immobilized FSC was 55 °C, which was 5 °C higher than that of the free enzyme, whereas its optimal pH was increased from 8.0 to 9.0; this indicates that the immobilized FSC had improved pH and thermal stability. After repeated use for 10 cycles, the activity of the immobilized FSC remained at more than 50%; after being stored at 4 °C for 30 days, its activity was still approximately 88%. We also found that the Km of the immobilized FSC was higher than that of the free enzyme. These results indicate that the performance of FSC was improved after immobilization, which is an important basis for the subsequent application of FSC in industrial production.

**Keywords:** immobilization; cutinase; chitosan; genipin

## 1. Introduction

Cutinases (EC 3.1.1.74) are serine esterases and are members of the α/β hydrolase family [1]. Cutinases are multifunctional hydrolases that can catalyze the hydrolysis of ester bonds of plant cutin [2] and various synthetic polyesters, such as poly($\varepsilon$-caprolactone) [3], polyethylene terephthalate [4], and poly(butylene succinate) [5]. In addition, cutinases can be used in organic media to improve the biotransformation efficiency of hydrophobic substrates such as triglycerides and short-chain fatty acids [3,6]. Nevertheless, the enzyme can become inactive when an organic solvent is present in the reaction medium because the intermolecular interactions between the enzyme molecules are changed [7]. Moreover, the stability of free cutinase is poor; thus, the enzyme can easily be inactivated by the operating conditions. Enzyme immobilization is an important approach to solve the above problems [1,8] and to promote the wider application of cutinase. Nikolaivits et al. immobilized cutinase from *Fusarium oxysporum* as crosslinked enzyme aggregates. The immobilized cutinase was used in isooctane for the synthesis of short-chain butyrate esters [9]. Sousa et al. used an approach based on the adsorption and establishment of affinity-like interactions with a biomimetic triazine-scaffolded ligand to immobilize cutinase from *F. solani*. This ligand was able to strongly bind cutinase and led to impressive half-lives of more than 8 h at 70 °C and of approximately 34 h at 60 °C for the bound protein [10].

Enzyme immobilization is a mature technology as it can be used to achieve the highest possible catalytic activity of a biocatalyst [11]. When enzymes are immobilized onto or into a solid medium, their thermal and operating stability can increase, which is beneficial for downstream processing [12]. Several natural polymers are used as carrier materials in the immobilization of enzymes. Among them is chitosan, which is one of the most favorable carrier materials for enzyme immobilization, attributable to its effortlessly adaptable chemical structure, in addition to many other outstanding properties, such as low

cost, easy obtainability, non-poisonousness, biodegradability, fungicidal and bacteriostatic properties, high mechanical strength, hydrophilicity, and high stability [13,14].

Genipin, a natural molecule purified from the fruits of *Genipa americana* and *Gardenia jasminoides* Ellis, is a kind of iridoid terpenoid compound that contains many active functional groups, such as carboxyl groups and hydroxyl groups [15]. It has been proven to be an appealing crosslinker for chitosan and amine-containing residues in the protein surface [16,17]. There are several different explanations for the immobilization mechanism of genipin. Nickerson et al. proposed that the ester group and olefin carbon atoms on genipin can react with the intramolecular and intermolecular amino groups of the amino-containing polymer. It may form a network structure [18]. Tsai et al. proposed that the mechanism of genipin crosslinking in amino polymers is a pH-dependent mechanism. In an acidic solution or a neutral solution, the C-3 alkene carbon atom of genipin has an affinity with the amino group on the amino-containing polymer. The dihydropyran ring opens to form a heterocyclic amine, which can form a polymer with a network structure. In an alkaline solution, genipin is nucleophilically attacked by the OH- and then opens the ring to form an aldehyde intermediate, which can undergo aldol condensation to obtain a macromolecular genipin polymer. The amino group on the amino-containing polymer and the terminal aldehyde group of the polymerized macromolecular genipin polymer form a polymer with a crosslinked network structure [19]. Though genipin has been applied to immobilize some enzymes, there are only a few reports on its use for cutinase immobilization.

In this work, magnetic chitosan beads were prepared as carriers, and purified *Fusarium solani* cutinase (FSC) was immobilized onto the beads, using genipin as the crosslinker. The immobilization conditions and the characteristics of the immobilized enzyme were investigated. Various properties, including pH stability, thermal stability, storage stability, and reusability, of the immobilized FSC were compared with those of the free enzyme. The immobilized FSC can be utilized in the biotransformation of hydrophobic substrates or the synthesis and degradation of some synthetic polyesters. The results presented in this study can be used as a sound basis for further applications of immobilized FSC.

## 2. Results and Discussion

*2.1. Effects of Immobilization Conditions*

2.1.1. Effect of Genipin Concentration

In general, the crosslinker concentration plays an important role in the combination between the enzyme and the carrier [20].The crosslinker can increase the degree of crosslinking within a certain range of concentrations.

Figure 1a illustrates the effect of genipin concentration on the activity of immobilized FSC. When the concentration of genipin was increased from 0.2 g/L to 0.6 g/L, the relative activity of the immobilized FSC showed an increasing trend. It appears that a low concentration of genipin is not sufficient for immobilization, which can in turn affect the binding efficiency of the enzyme. However, with an increasing genipin concentration, the binding amount can increase and the binding efficiency of the enzyme can in turn be improved. When the genipin concentrations were 0.6 g/L and higher, the relative activity of immobilized FSC decreased with increasing genipin concentration. This may have been due to parallel reactions between genipin molecules when their concentrations were high. This can reduce the reaction between groups on the surface of chitosan during enzyme immobilization [21].

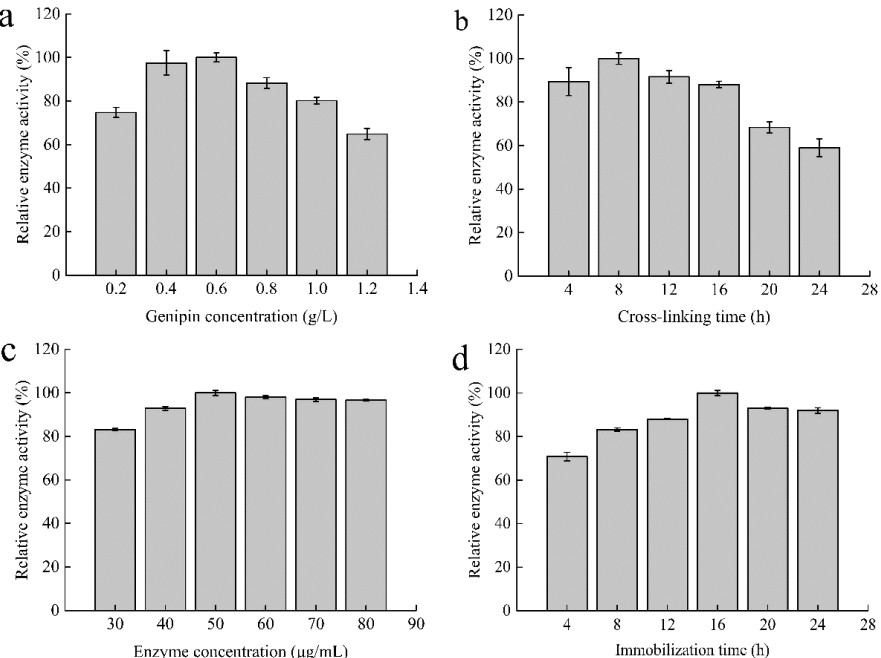

**Figure 1.** Effects of genipin concentration (**a**), crosslinking time (**b**), enzyme concentration (**c**), and immobilization time (**d**) on immobilized FSC activity.

### 2.1.2. Effect of Crosslinking Time

Figure 1b shows the change in the relative activity of immobilized FSC with increasing crosslinking time from 4 h to 24 h. When the crosslinking time was 8 h, the relative activity of the immobilized FSC reached its maximum value. This indicates that the number and size of the space grid in the carrier under this crosslinking reaction time were at their optimal values, thus allowing optimal adsorption of enzyme molecules onto chitosan beads [22]. In general, the space grid structure of the carrier increases with increasing crosslinking time, which can in turn restrict the entry of more enzyme molecules [23]. For this reason, when the crosslinking time exceeded 8 h, the relative activity of the immobilized enzyme showed a downward trend.

### 2.1.3. Effect of Enzyme Concentration

After crosslinking, there is a sufficient amount of active free radicals on the carrier. This experiment was conducted to adjust the amount of enzyme to ensure not only that no enzymes are wasted, but also that there are as many immobilized enzymes as possible. Figure 1c shows that the relative activity of immobilized FSC reached the maximum when the enzyme concentration was 50 μg/mL. Further increasing the enzyme concentration led to a slight decrease in relative activity, rather than an obvious increase, which may have been caused by the overload and space hindrance of the carrier [22,24].

### 2.1.4. Effect of Immobilization Time

Immobilization time is another vital variable affecting the enzyme immobilization process. Figure 1d shows that the relative activity of the immobilized FSC increased with extended immobilization time from 4 h to 16 h, which indicates that the enzyme molecules need adequate time to enter into the space grid structure of the carrier in order to become immobilized. However, when the immobilization time is too long, the enzyme molecules on the carrier surface can become too crowded, and their contact with the substrate can in turn be restricted [25]. For this reason, the relative activity of the immobilized enzyme was reduced when the immobilization time exceeded 16 h. In addition, the enzyme can be easily inactivated when it is placed at room temperature for a long period of time. This may be another reason that the relative activity decreased.

In summary, the optimum conditions for the immobilization of FSC were as follows: genipin concentration, 0.6 g/L; crosslinking time, 8 h; enzyme concentration, 50 μg/mL; and immobilization time, 16 h.

### 2.2. Characterization of Magnetic Genipin-Crosslinked Chitosan Beads

Figure 2a shows the prepared magnetic chitosan beads. The beads have a uniform diameter of approximately 0.5 mm and are spherical. Figure 2b shows an overview of the structure, Figure 2c shows the surface structure, and Figure 2d shows the internal structure. The beads have a spherical structure (Figure 2b); their surfaces are uneven and porous and have large pore sizes (Figure 2c), which can decrease the mass transfer resistance between the substrate and the product during the immobilization process. This has a beneficial effect on enzyme immobilization. The interior of the beads is crosslinked and contains many pores (Figure 2d).

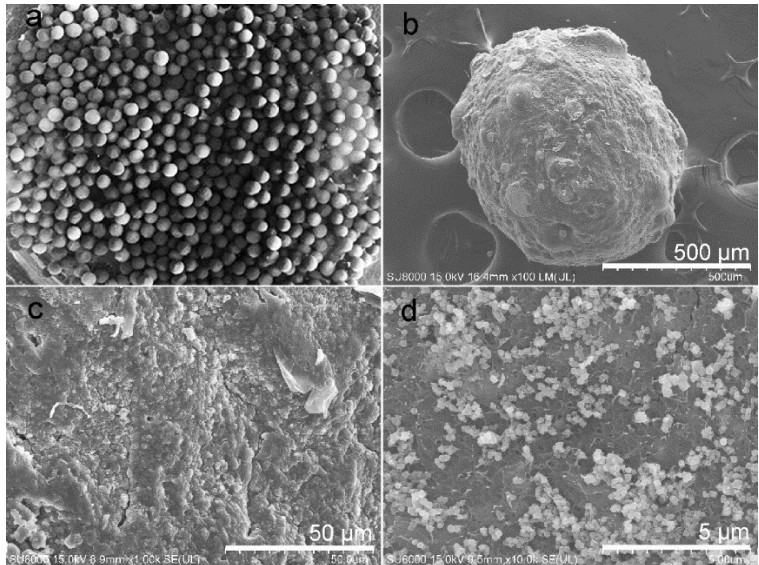

**Figure 2.** Micrograph of magnetic genipin-crosslinked chitosan bead. Digital photograph (**a**); whole structure (SEM) (**b**); surface structure (SEM) (**c**); internal structure (SEM) (**d**).

The FTIR spectra of $Fe_3O_4$, chitosan, and magnetic genipin-crosslinked chitosan beads are shown in Figure S1. The magnetic genipin-crosslinked chitosan beads could retain the characteristic absorption peaks of chitosan and $Fe_3O_4$; besides these peaks, a new peak of C=O appeared at 1638 cm$^{-1}$, which was an indication that the crosslinking reaction between genipin and chitosan had occurred.

### 2.3. Effects of Reaction pH and Temperature on FSC Activity

As shown in Figure 3a, the optimal pH of free FSC was 8.0, whereas that of immobilized FSC was higher, at pH 9.0. The relative activity in the acidic region of immobilized FSC was obviously lower than that of free FSC; conversely, the relative activity in the alkaline region of the immobilized enzyme was higher than that of the free enzyme. This may be attributed to the cationic property of chitosan during salt formation and the acidity of the product, p-NP; moreover, the immobilized enzyme may also require an alkaline pH in order to exhibit catalytic activity [26]. Another possible reason for the decrease in the relative activity of the immobilized enzyme at acidic pH is that the interaction between the residual charge of activated chitosan and the coupling enzyme may have an adverse effect on the active site. Similar results have been previously observed in some other enzymes [27,28].

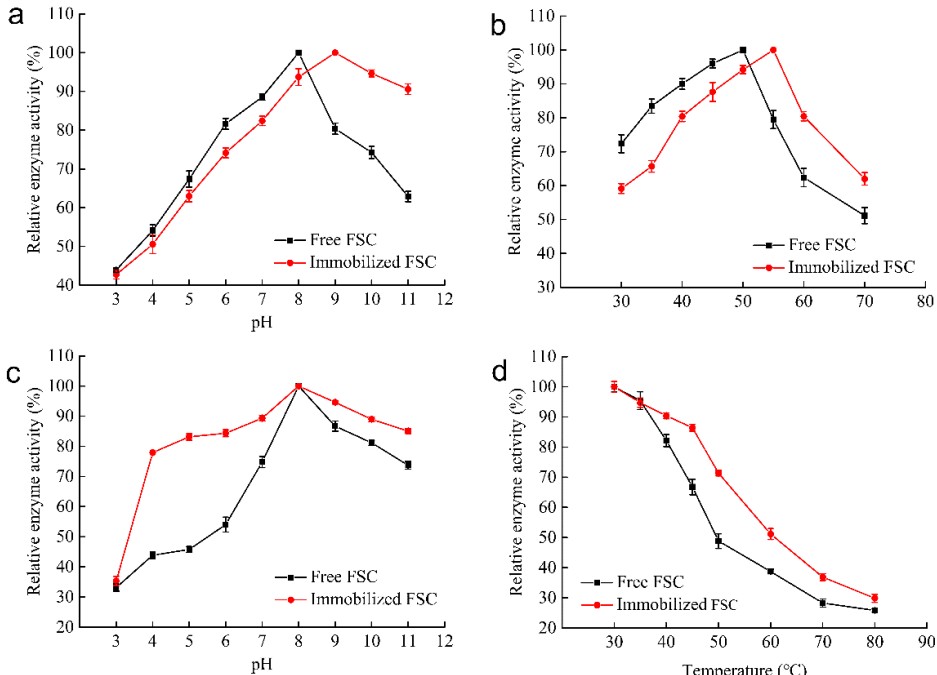

**Figure 3.** Effects of pH (**a**) and temperature (**b**) on the activity of free FSC and immobilized FSC and effect of pH (**c**) and temperature (**d**) on the stability of free FSC and immobilized FSC.

As can be seen from Figure 3b, the optimal temperature of free FSC was 50 °C, while that of immobilized FSC was 55 °C. This indicates that the immobilization, by enhancing the structural rigidity of the protein, enhances its resistance to increased temperatures. Furthermore, the relative activity of immobilized FSC at 60 °C, 70 °C, and 80 °C was significantly lower than that of free FSC. This may be due to the fact that the thermal stability of the carrier was poor. This also indicates that, after being immobilized, the enzyme activity becomes more dependent on temperature [21].

The optimum pH for both immobilized and free FSC was 8.0; however, the pH stability of the immobilized FSC was obviously higher than that of free FSC (Figure 3c). The relative activity of immobilized FSC remained at more than 80% in the pH range of 5.0 to 11.0. When the pH was less than 8.0, the relative activity of the free enzyme decreased sharply with decreasing pH value, which may have been due to the partial inactivation of the enzyme. The improved stability of the immobilized enzyme under acidic conditions is likely related to the intramolecular salt bridge formed between the anionic groups on the enzyme and the cationic groups on the chitosan carrier [29]. Compared with that of free FSC, the stability of the immobilized FSC was improved in the tested pH range, which indicates that the immobilization can not only maintain the activity of the enzyme but also improve its stability.

To determine their thermal stability, the immobilized FSC and free FSC were incubated at various temperatures between 30 and 80 °C for 3 h, after which their residual activities were assayed. As shown in Figure 3d, the relative activity of both immobilized FSC and free FSC decreased with increasing temperature from 30 to 80 °C. After incubation at 50 °C for 3 h, the relative activity of immobilized FSC remained at more than 70%, while that of the free enzyme decreased to 50%. This indicates that the immobilized FSC had higher thermal stability than the free FSC. This generates a relatively stable spatial framework that enhances the thermal stability of the enzyme molecule [24], thereby reducing the rate of enzyme inactivation at high temperatures.

### 2.4. Reusability of Immobilized FSC

As can be seen in Figure 4, the relative activity of immobilized FSC was more than 80% after being reused for three cycles and was more than 60% after being reused for six cycles.

With the increasing number of reuse cycles, the relative activity of immobilized FSC decreased, reaching a value of 50% of its original activity after being reused for 10 cycles. This result shows that the FSC immobilized onto magnetic genipin-crosslinked chitosan beads has good reusability. The loss of enzyme activity can be clearly attributed to repeated use, which can cause a change in the molecular conformation, enzyme inactivation, and weakening of the binding strength between the carrier matrix and the enzyme molecule [20].

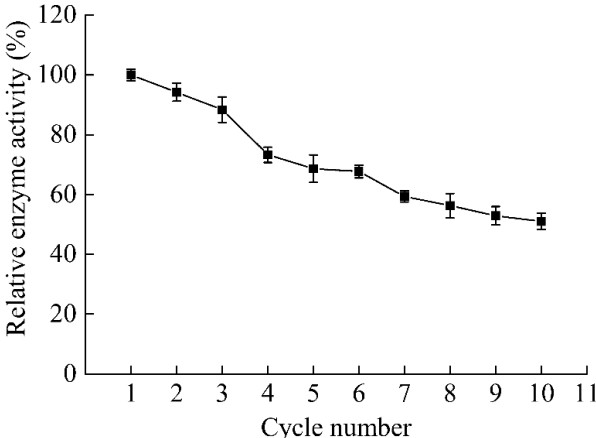

**Figure 4.** Reusability of immobilized FSC.

### 2.5. Storage Stability of FSC

The relative activity of both the free and the immobilized FSC lessened after being stored at 4 °C for one month; however, the storage stability of the immobilized FSC was significantly higher than that of the free FSC (Figure 5). This may be due to the immobilization on the carrier, which can protect the enzyme so that its molecular conformation is not easily changed. The residual relative activity of immobilized FSC was higher than 95% after 10 days of storage, while it remained above 88% after 30 days. These results indicate that the storage stability of the immobilized FSC is significantly higher than that of the free FSC; thus, immobilization can clearly increase the storage stability of the enzyme, extending its preservation time so that it can be suitably be applied in some long-term applications.

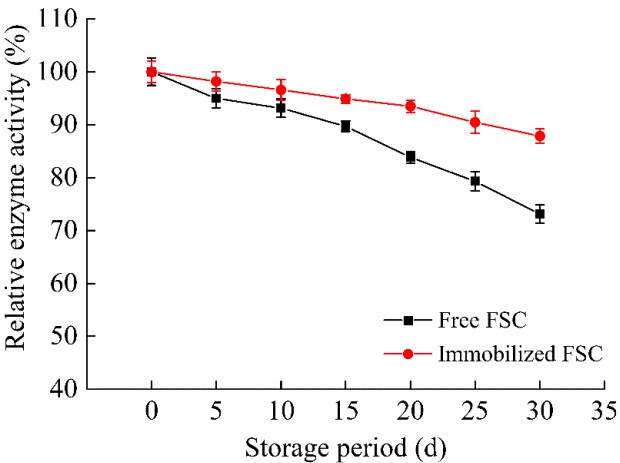

**Figure 5.** Storage stability of free FSC and immobilized FSC.

### 2.6. Kinetic Parameters

To determine the relationship between the substrate and the activity of FSC, Lineweaver–Burk plots (Figure S2) were constructed by plotting $1/V$ against $1/[S]$ using the double reciprocal plotting method, and the kinetic parameters ($K_m$ and $V_{max}$) of the enzymatic reaction were calculated (Table 1). The $K_m$ is an indicator of the affinity of an enzyme

towards its substrate: the higher the $K_m$ value, the lower the affinity of an enzyme towards its substrate [30]. The $K_m$ of immobilized FSC was greater than that of free FSC, which indicates that immobilization can reduce the affinity of the enzyme towards its substrate. Although the adaptability of an enzyme to various environmental conditions can be improved by immobilization, it is inevitable that the $K_m$ value will increase because immobilization can reduce the flexibility of the binding between the substrate and the enzyme. The presence of the carrier increases the steric hindrance between the substrate and enzyme activity site and also increases the limits of diffusion between the substrate and the product [13,31]. The $V_{max}$ value of immobilized FSC was lower than that of free FSC. This is because the free FSC can be evenly dispersed in the substrate, but the immobilized FSC is bound to the space grid structure of the carrier, causing its probability of being in contact with the substrate to be lower than that of free FSC; as a result, the immobilized FSC had a lower $V_{max}$ value [32].

**Table 1.** Kinetic parameters of free and immobilized FSC.

| Kinetic Parameters | Free FSC | Immobilized FSC |
|---|---|---|
| $K_m$ (mM) | 10.3 | 14.4 |
| $V_{max}$ ($\mu$M·min$^{-1}$) | 178.6 | 140.8 |

## 3. Materials and Methods

### 3.1. Materials

*Fusarium solani* cutinase (FSC) was prepared from the fermentation broth of recombinant *Pichia pastoris*; the detailed preparation processes have been reported in our previous work [33]. Chitosan (degree of deacetylation = 85–95%), $Fe_3O_4$ nanoparticles (particle size = 100 nm), and p-nitrophenyl butyrate (p-NPB) were obtained from Aladdin Biochemical Company (Shanghai, China). Genipin ($\geq$98% purity) was obtained from Linchuan Zhixin Biotechnology Co., Ltd. (Fuzhou, China). All other reagents and chemicals used were of the highest purity available.

### 3.2. Immobilization of FSC onto Magnetic Chitosan Beads

Three percent chitosan in 2% aqueous acetic acid was mixed with $Fe_3O_4$ nanoparticles; after this, 20 mL of the mixture was dropwise added into 100 mL of kerosene containing 1% Span 80 and 0.3% Tween80. The mixture was stirred at 350 rpm for 1 h to prepare a homogeneous W/O magnetic chitosan emulsion, which was then quickly poured into a coagulant (2M NaOH: anhydrous ethanol = 4:1, *v/v*); then, the mixture was stirred for another 30 min. After being allowed to stand overnight, the dispersed phase on the surface of the magnetic chitosan beads was washed with anhydrous ethanol, followed by repeated washing with deionized water; then, the magnetic chitosan beads were stored in deionized water at 4 °C thereafter.

Two grams of magnetic chitosan beads were incubated with 20 mL of genipin solution at different concentrations (0.2, 0.4, 0.6, 0.8, 1.0, and 1.2 g/L), and the crosslinking reaction was allowed to take place at 55 °C for varying time periods (4, 8, 12, 16, 20, and 24 h). After washing with deionized water, the beads were kept at 4 °C in deionized water.

Two grams of magnetic genipin-crosslinked chitosan beads were added to 40 mL of 0.1 M boric acid buffer (pH 8.0) containing various amounts of FSC (30, 40, 50, 60, 70, and 80 $\mu$g/mL), and the immobilization was executed at room temperature for different times (4, 8, 12, 16, 20, and 24 h). Subsequently, the beads were washed with 50 mM sodium phosphate buffer (pH 8.0) to remove uncombined FSC. Finally, the obtained immobilized enzyme was stored in the same buffer at 4 °C until further use.

### 3.3. Characterization of Magnetic Genipin-Crosslinked Chitosan Beads

3.3.1. Scanning Electron Microscopy (SEM)

The surface and the interior of the magnetic genipin-crosslinked chitosan beads were observed by SEM (XL30 ESEM-FEG, FEI, Eindhoven, The Netherlands) operated at an accelerating voltage of 15 kV. Before the measurement, each sample was coated with a thin layer of gold.

3.3.2. Fourier Transform Infrared (FTIR) Spectroscopy

FTIR spectra of chitosan and magnetic genipin-crosslinked chitosan beads were determined using a Cary 660 FTIR spectrometer (Agilent Technologies, Inc., Santa Clara, CA, USA). The samples were mixed with KBr powder and pressed into pellets before being subjected to the FTIR measurement, which was carried out at a frequency range of 4000 to 400 $cm^{-1}$.

### 3.4. Analysis of Enzyme Activity and Protein Concentration

The enzyme activity was assayed according to the p-nitrophenol method, with modifications [34]. First, 100 μL of free enzyme or 10 mg of immobilized enzyme was added to 1380 μL of 50 mM sodium phosphate buffer (pH 7.0), equilibrated to 37 °C, and 20 μL of 100 mM p-NPB in acetonitrile was added. After 10 min, 200 μL of 0.5 mol/L trichloroacetic acid solution was added and mixed to terminate the reaction. Then, the mixture was added to 200 μL of 0.5 mol/L sodium carbonate solution and centrifuged at 12,000 r/min for 1 min. Production of p-nitrophenol was measured at 405 nm using a recording spectrophotometer. All assays were performed in triplicate. Protein concentration was estimated using the BCA Protein Assay Kit (Sangon Biotech Co., Ltd., Shanghai, China), using bovine serum albumin as a standard.

### 3.5. Effects of pH and Temperature on Enzyme Stability

(1) Optimization of pH and temperature: The activities of free and immobilized FSC were estimated at different pH levels from 3.0 to 11.0 and at different temperatures from 30 to 80 °C.

(2) Determination of enzyme stability at different pH levels: Both free and immobilized FSC were incubated at different pH levels (from 3.0 to 11.0) for 24 h at 30 °C, and their activities were determined.

(3) Determination of thermostability of enzyme: Both free and immobilized FSC were dissolved in potassium phosphate buffer (pH 8.0) and then stored at various temperatures from 30 to 80 °C for 3 h.

The residual activity was measured according to the methods explained in Section 2.4. The enzyme activity assayed under the optimal conditions was considered as 100%.

### 3.6. Storage Stability and Reusability

The immobilized FSC was reused for 10 cycles, and the enzyme activity after each cycle was assayed. After each cycle, the immobilized FSC was filtered, cleaned with 50 mM sodium phosphate buffer (pH 8.0), and then mixed with fresh reaction substrate. The activity of the immobilized FSC in the first cycle was defined as 100%, and the activity in later cycles was expressed as relative enzyme activity (in relative to the first cycle).

The storage stability was assayed by keeping the free or immobilized FSC with the same activity in 50 mM sodium phosphate buffer (pH 8.0) at 4 °C for 1 month. During this time, their enzyme activities were measured at days 0, 5, 10, 15, 20, 25, and 30. The FSC activity on day 0 was defined as 100%, and the activities on other days were expressed as relative enzyme activity (relative to that of day 0).

### 3.7. Determination of Kinetic Parameters

$K_m$ and $V_{max}$ values were determined by calculating the initial reaction velocity of free and immobilized FSC, determined using various concentrations of pNPB (the substrate) in 50 mM sodium phosphate buffer (pH 8.0) at 37 °C.

### 3.8. Statistical Analysis

All experiments and enzyme assays were executed in triplicate, and the results are presented as mean ± S.E. (standard error). All computations were performed by employing the statistical software SPSS (version 22.0).

### 4. Conclusions

In this work, magnetic genipin-crosslinked chitosan beads were prepared and characterized, and then used to immobilize FSC. The optimal preparation conditions for immobilized FSC were as follows: genipin concentration, 0.6 g/L; crosslinking time, 8 h; enzyme concentration, 50 μg/mL; and immobilization time, 16 h. Compared with free FSC, the immobilized FSC had higher pH and thermal stability. The immobilized FSC also exhibited outstanding recyclability as it could maintain >50% residual activity after 10 cycles. It also retained approximately 90% of its activity after being stored at 4 °C for 30 days. These extraordinary properties render the immobilized FSC a potential candidate for commercial applications.

**Supplementary Materials:** The following are available online at https://www.mdpi.com/article/10.3390/catal11101158/s1, Figure S1: FTIR, Figure S2: Lineweaver–Burk plots.

**Author Contributions:** Conceptualization, Z.W.; methodology, T.S.; validation, J.Z.; formal analysis, Z.W.; resources, Z.W.; data curation, T.S.; writing—original draft preparation, Z.W.; writing—review and editing, T.S.; visualization, J.Z.; project administration, Z.W.; funding acquisition, Z.W. All authors have read and agreed to the published version of the manuscript.

**Funding:** This research was funded by the National Natural Science Foundation of China, grant number 31570097, LiaoNing Revitalization Talents Program, grant number XLYC1807034 and the Talent Program of Shenyang Agricultural University, grant number 2021Y001.

**Conflicts of Interest:** The authors declare no conflict of interest.

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
