# Peer review of "Immobilization of Fusarium solani Cutinase onto Magnetic Genipin-Crosslinked Chitosan Beads"

_catalysts, doi:10.3390/catal11101158_

Round 1

Reviewer 1 Report

I found presented work interseting and well prepared. 

Statistical analysis - please provide the programme you used to data analysis 

Table 1. Vmax (μM·min) - Are Authors sure about the unit?

Author Response

Thanks to the reviewer for your time and thoughtful comments again. Below are our responses to your comments.

1. Statistical analysis - please provide the programme you used to data analysis 

The software information was provided in the Section 3.8.

2. Table 1. Vmax (μM·min) - Are Authors sure about the unit?

It is our mistake. The unit in Table 1 was revised.

Reviewer 2 Report

The authors present here a new immobilization method for a cutinase, with genepin crosslinking in nanoparticles. Their research is sound, and the methods used accurate. Only minor editing on the English to enhance the readability of the manuscript and some questions should be answered.

Abstract: A higher km actually would mean the enzyme is less specific for the substrate.

Line 25: Cutinases belong the hydrolase family, calling them lyases might cause confusion.

Line 27: “can be used in organic media” rather than “can use organic solvents as reaction media”

Line 35: “Enzyme immobilization is a mature technology…”

Line 47: “amine containing residues in the protein surface”.

Line 62: Reword for clarity. The sentence is repetitive and unclear.

Line 68: It might be that the available genipin is not enough for immobilization, rather than a problem of insufficient binding.

Figure 1: It would be good if the authors could also show the immobilization yield (immobilized/offered protein) to the graph, thus further supporting their claims on the previous paragraphs. Also when the authors refer to relative enzyme activity, do they mean compared to the offered activity?

Line 120-132: I think it should be shortened, highlighting the presence of the peak that supports the genipin crosslinking and leaving all the other out. It reads to repetitive and gives no real information to the reader. I would probably move the graph to supplementary information and just add a comment on the appearance of the peak at 1638 cm-1.

Line 143: Delete this sentence, as the explanation is much clearer in the next sentences and feels repetitive otherwise.

Line 157: Reword for clarity. This indicates that the immobilization, by enhancing the structural rigidity of the protein, enhances its resistance to increased temperature.

Line 160: Poor thermal stability of the carrier rather than the much more complicated sentence.

Line 168: May be due to the partial inactivation of the enzyme….

Line 181: The statement “After the enzyme was immobilized..” is not informative at all. It does not give any new information or rationale for the increased stability.

Line 191: Reusability: have the authors seen any  leaching of the enzyme to the bulk of the reaction? This could also explain the decreased activity over time.

Line 216: it might be good to add the graph of the plot, even in the SI.

Reviewer 3 Report

Developing active immobilized enzymes and characterization of their use conditions is critically important, but the immobilization of cutinase has been studied before. 

- solani cutinase FsC has been immobilized on a triazine-based affinity ligand, which was synthesized in agarose. The immobilized catalyst showed an impressive 57-fold increase of t1/2 at 60 ◦C, compared to the free enzyme - Sousa, I.T.; Lourenço, N.M.T.; Afonso, C.A.M.; Taipa, M.A. Protein stabilization with a dipeptide-mimic triazine-scaffolded synthetic affinity ligand. J. Mol. Recognit. 2013, 26, 104–112

- a recombinant F. oxysporum cutinase was  immobilized following CLEA methodology using the crude intracellular fraction of the E.coli host. Its thermostability increased 3-fold at 35 ◦C, and it could be reused for the synthesis of butyl butyrate up to four times, with less than 50% reduction of yield - Nikolaivits, E.; Makris, G.; Topakas, E. Immobilization of a cutinase from Fusarium oxysporum and application in pineapple flavor synthesis. J. Agric. Food Chem. 2017, 65, 3505–3511

Please improve the Introduction section by presenting other results for the immobilization of cutinase. A comparation between other results and yours is necessary. 

Please explain the novelty of your study, its advantages compared to others.

Please add information regarding the immobilization mecanism. How is genipin influencing the immobilization mechanism? The cost of genipin - increase in the cost of the immobilization process - can be justify taking into account only slightly changes in operation parameters?

Figure 4 is ambiguous and confusing. Please present the results by means of the enzyme activity obtained and not as  relative to the first cycle. 

Please present in more detail the methods used in the enzyme activity assay. 

Solvent tolerance is an important property of an immobilized enzyme as it allows medium engineering to optimize performance characteristics such as enantioselectivity. An analyse of tolerance towards solvents of different polarity would improve your article.

The evaluation of immobilized cutinase selectivity would be interesting.

The analysis of t1/2 would be interesting.

Round 2

Reviewer 3 Report

The article can be accepted.